# A Rare Neuro-Ophthalmological Condition in a Patient with Lung Adenocarcinoma: The Eight-and-a-Half Syndrome, Case Report and Review of the Literature

**DOI:** 10.3390/brainsci12040451

**Published:** 2022-03-28

**Authors:** Alessandro Cruciani, Francesco Motolese, Antonio Todisco, Vincenzo Di Lazzaro, Giovanni Assenza

**Affiliations:** Neurology, Neurophysiology and Neurobiology Unit, Department of Medicine, Università Campus Bio-Medico di Roma, 00128 Rome, Italy; f.motolese@unicampus.it (F.M.); antonio.todisco@unicampus.it (A.T.); v.dilazzaro@unicampus.it (V.D.L.); g.assenza@unicampus.it (G.A.)

**Keywords:** eight-and-a-half syndrome, neuro-ophthalmology, neuro-oncology

## Abstract

The eight-and-a-half syndrome is a rare neuro-ophthalmological condition caused by a structural lesion in the dorsal portion of the pons, involving critical areas of the brainstem, i.e., medial longitudinal fasciculus (MLF), abducens nucleus, facial genu, and colliculus. It is characterized by internuclear ophthalmoplegia with horizontal gaze palsy and peripheral facial palsy. Although the syndrome is most frequently caused by vascular or demyelinating diseases, several different underlying causes might occur. Herein, we describe a case of the eight-and-a-half syndrome caused by a lung adenocarcinoma metastasis localized in the lower pontine tegmentum. Then, we review the current literature on the underlying causes of the eight-and-a-half syndrome.

## 1. Introduction

Diseases of the brainstem represent a challenge even for most expert clinicians. This peculiar part of the nervous system is densely packed in both grey and white matter structures. In the brainstem are located multiple nuclei subserving a vast array of functions, including oculomotion and consciousness, and there are several ascending and descending neural pathways. Not surprisingly, even a small lesion might cause a dramatic clinical picture because of the complex anatomical nature of the region [1]. Once the anatomical localization of the lesion is recognized, there could be several underlying causes leading to brainstem damage, with each one leading to a different diagnostic and therapeutic scenario [2]. The eight-and-a-half syndrome is a rare condition caused by a lesion located in the dorsal part of the pons involving the pontine reticular formation, the medial longitudinal fasciculus, the facial genu, and the colliculus [3]. This characteristic and rare localization leads to a combination of unique clinical findings characterized by ipsilateral conjugate horizontal gaze palsy and limited adduction of the ipsilateral eye, i.e., the one-and-a-half-syndrome, plus peripheral facial nerve palsy [4]. Most commonly, it is caused by cerebrovascular diseases [5], but occasionally, it could be caused by demyelinating disorders [6], meningitis, or brain metastases [7,8].

Herein, we describe a case of the eight-and-a-half syndrome caused by a lung adenocarcinoma metastasis localized in the lower pontine tegmentum. Then, we review the available literature to focus the attention on the causes underlying the eight-and-a-half syndrome.

## 2. Case Report

A 56-year-old man with no past medical history presented at the emergency department because of subacute dyspnea. A chest computed tomography scan (CT scan) was performed, showing subpleural lung nodules in the right upper lobe with massive pleural effusion. A transbronchial biopsy specimen obtained from the lesion revealed malignant cells suggestive of adenocarcinoma. The patient started chemotherapy with carboplatin (CBDCA regimen) and PD-1 inhibitors (pembrolizumab). Two months after starting chemotherapy, the patient returned to the emergency department, complaining of right facial drooping and diplopia. These symptoms appeared about two weeks earlier and were slowly getting worse at the time of presentation. The neurological examination showed right facial palsy with a peripheral pattern, rightward conjugate gaze palsy, and right internuclear ophthalmoplegia (impaired right eye adduction with nystagmus in the left eye). The leftward gaze evoked left-beating nystagmus. The patient could not abduct the right eye nor adduct the left eye, even with the closure of the other eye, while convergence was spared. Magnetic resonance imaging (MRI) showed multiple ring-enhancing lesions in the cerebellum, the temporo-occipital lobe, and the dorsal tegmentum of the pons (Figure 1). These lesions were highly suggestive of metastatic lesions. The patient started a radiotherapy treatment of the brainstem lesion and was then discharged at home. Two months later, the patient died of cardiopulmonary complications.

## 3. Discussion

The combination of the one-and-a-half syndrome, i.e., conjugate horizontal gaze palsy and limited adduction of the ipsilateral eye, plus peripheral facial nerve palsy constitutes the so-called “eight-and-a-half syndrome” [9,10]. This is a rare neuro-ophthalmological condition caused by lesions involving the pontine reticular formation, the medial longitudinal fasciculus, the facial genu, and the colliculus [3,10].

Eye movements are controlled by three different cranial nerves III (or oculomotor), IV (or trochlear), and VI (or abducens). The generation of ocular movements is due to the complex integration of these three nerves. The abducens nucleus is set below the fourth ventricle, localized in the portion of the brainstem between the pons and the medulla, surrounded by the genu of the facial nerve [11]. Its function is to abduct, i.e., move laterally, the eye. The abducens nucleus contains the motoneurons innervating the muscle fibers of the lateral rectus muscle, interneurons, and other neurons projecting to the contralateral third cranial nerve nucleus—constituting the medial longitudinal fasciculus (MLF) [11]. The trochlear nerve has its nucleus just below the aqueduct in the mesencephalon near the central periaqueductal grey matter [12]. It contains neurons that are directed towards contralateral superior oblique muscle. The main action of the trochlear nerve is to depress and intort the eyeball. The oculomotor nucleus is contiguous to the trochlear nucleus, localized rostrally at the ventral border of the periaqueductal grey matter and extended to the posterior commissure [12]. The fibers that radiate from the oculomotor nucleus reach the ipsilateral medial rectus, inferior rectus, inferior oblique muscles, and the contralateral superior rectus muscle.

The horizontal gaze is due to the action of the medial and lateral rectus, innervated by the oculomotor and abducens nerve, respectively. When the VI cranial nerve activates the lateral rectus muscle of the homolateral eye, it also induces the activation of the medial rectus muscle of the contralateral eye through fibers running in the MLF. These structures—the third and sixth nuclei and MLFs—are controlled by the pontine reticular formation that integrates afferences from different parts of the brain, including vestibular inputs and cerebellar projections. Thus, the core of the horizontal eye control is located in the pons, specifically in the paramedian pontine reticular formation (PPRF) that contains the excitatory and inhibitory burst neurons for horizontal gaze [13] and in the nucleus raphe interpositus (RIP) containing the omnipause neurons [14]. The activation of these groups of neurons allows conjugated movements of the two eyes.

The vertical and torsional gaze is due to the action of the oculomotor and trochlear nerves. The generation of vertical movements occurs in the mesencephalic region, precisely in the mesencephalic reticular formation (MRF) [15] that contains the rostral interstitial nucleus of the medial longitudinal fasciculus (RIMLF), the M-group (M), and interstitial nucleus of Cajal (INC) [16]. These structures control the vertical gaze and some muscles of the eyelid. The RIMLF contains the burst excitatory neurons, while the INC contains the burst inhibitory neurons.

As previously mentioned, the MLF is not only a bundle of fibers connecting the abducens nucleus to the contralateral third nerve nucleus, but it also receives projections from the vestibulocochlear nerve. The MLF starts in the rostral interstitial nucleus of the medial longitudinal fasciculus (riMLF) situated in the mesencephalic reticular formation. Then, it reaches the nucleus of cranial nerve III and then the nuclei of cranial nerves IV and VI [17].

In our patient, the metastasis was located in the lower tegmentum of the pons, thus damaging the abducens nucleus, the MLF, and the nerve fibers coming from the seventh nerve nucleus. The eight-and-a-half syndrome was due to the damage of these structures in the pons.

Lung cancer is the most common cause of brain metastases (BMs), and about 10% to 36% of patients with lung cancer develop BMs during the course of the disease [18]. Although the preferential sites of adenocarcinoma BMs are the cerebellum and the distal middle cerebral artery territory [19], some cases of brainstem metastases are described [20]. The treatment of brainstem metastases is particularly challenging. Some authors have suggested the application of the gamma knife surgical approach coupled to whole-brain radiation therapy, but the results are controversial [21].

Due to its clinical characteristics, the eight-and-a-half syndrome should be differentiated from other brainstem syndromes, which typically cause ipsilateral cranial nerve lesions and contralateral long tract signs. In particular, the Marie–Foix syndrome (lateral pontine syndrome) is characterized by contralateral symptoms—hemiplegia and loss of temperature and pain sensation—and homolateral symptoms (limb ataxia, facial weakness, loss of pain and temperature sensation of the face, nystagmus, and hearing loss). In most cases, it occurs after the occlusion of perforating branches of the basilar and anterior inferior cerebellar arteries [22]. The Foville syndrome (inferior medial pontine syndrome) is characterized by contralateral hemiparesis or hemiplegia, peripheral ipsilateral facial nerve palsy, and conjugate gaze palsy with the inability to look to the side of the lesion. Often it is due to the occlusion of the paramedian branches of the basilar artery [1]. Finally, ventral pontine syndromes are associated with different clinical conditions such as the Raymond syndrome and the Millard–Gubler syndrome. In the former, a unilateral lesion of the ventromedial pons causes lateral gaze weakness, due to the ipsilateral lateral rectus paresis, and contralateral hemiplegia [23]. The Millard–Gubler syndrome is associated with ipsilateral abduction deficit, ipsilateral facial muscle weakness, and contralateral hemiparesis or hemiplegia. It is caused by lesions to the ventral aspect of the caudal pons [1]. However, the clinical picture associated with brainstem syndromes is pretty straightforward, and a careful clinical evaluation is mandatory for differential diagnoses. In our patient, the combination of symptoms was highly suggestive of the eight-and-a-half syndrome.

The eight-and-a-half syndrome was first described by Eggenberger et al. in the mid-nineties, which assessed the combination of these unique clinical findings in two patients with vertebral basilar stroke and one patient affected by giant cell arteritis [3]. However, this rare phenotype can result from a broad subset of causes. Our literature review analyzed 30 cases of the eight-and-a-half syndrome published since the first description by Eggenberger (Table 1).

The most frequent etiology is cerebrovascular disease (73%), mainly including ischemic stroke (60%) [24]. When a cerebrovascular etiology is suspected, a magnetic resonance angiography (MRA) might be useful to better evaluate possible stenosis or occlusion. Pontine hemorrhage due to cavernomas (7%) [25,26] is also reported. Maier et al. [25] described a sudden onset of the eight-and-a-half syndrome in a 24-week pregnant woman, presenting with the weakness and tingling of the left limbs, subsequent facial asymmetry, and progressive eye movement limitations. A single case in the literature due to intracranial capillary telangiectasia (3%) was reported by Li and colleagues [27]. They described the case of a patient who experienced sudden severe headache during sleep, followed by blurred vision and left facial hyposthenia lasting for 2 h. Xia et al. [28], instead, reported the occurrence of the eight-and-a-half syndrome in the context of a brainstem bleeding of unknown origin (3%).

This peculiar picture can be occasionally caused by demyelinating disorders (17%), which is the second main pathogenetic mechanism according to our review [2]. Mortzos and colleagues presented the case of a 12-year-old boy who developed a subacute double vision and a right facial palsy after 10 days of dizziness, nausea, and headache and later was diagnosed with childhood multiple sclerosis [29].

Although anecdotic, tuberculous meningitis and tuberculoma are reported as rare culprit lesions (7%) for the eight-and-a-half syndrome [7,30]. Lastly, a case of point-like brain metastasis spreading from a lung tumor is also described (3%) [8].

Every process involving the lower pontine tegmentum might cause this constellation of symptoms, causing clinical pictures similar to but slightly different from the eight-and-a-half syndrome. Xia and colleagues [28] proposed an innovative classification of the eight-and-a-half syndrome, taking as core characteristics the variability of the clinical presentation. They categorized the clinical spectrum into three types, namely classic eight-and-a-half syndrome, eight-and-a-half syndrome variants, and eight-and-a-half plus syndromes. Variants include cases of the eight-and-a-half syndrome associated with unilateral vertical gaze palsy [31] and bilateral horizontal gaze palsy [32].

Eight-and-a-half plus syndromes consist of complex clinical spectra, in which wider pontine lesions enrich the classical subset with ulterior unusual signs and symptoms. Nine syndromes (eight-and-a-half syndrome associated with hemiparesis or hemiataxia) [33,34] were first reported by Rosini et al. [33], who described a case of point-like acute pontine infarction, presenting with contralateral hemiparesis and hypesthesia of the left arm, due to a concomitant involvement of the corticospinal tract and medial lemniscus. The thirteen-and-a-half syndrome (eight-and-a-half syndrome associated with ipsilateral trigeminal nerve palsy) [35] and the fifteen-and-a-half syndrome (eight-and-a-half syndrome associated with bilateral facial nerve palsy) [36] were also reported. The former, described by Allbon and colleagues, was caused by a metastatic pulmonary lymphoma, in a 50-year-old male with a history of multiple kidney transplantations [35]. The latter, reported by Bae and colleagues, was characterized by bilateral facial hyposthenia, in addition to the classical one-and-a-half ocular limitation, as a clinical presentation of pontine stroke in a 67-year-old man [36]. Notably, a case of the ischemic nine syndrome regressed to a pure eight-and-a-half syndrome after a Tissue plasminogen activator (t-PA) treatment [37].

**Table 1 brainsci-12-00451-t001:** Literature review.

Etiopathology		Reference
Cerebrovascular disease (73%)	Ischemic (60%)	Ahmed [24]
		Almutlaq [38]
		Bocos-Portillo [9]
		Cole [37]
		Duffy [10]
		Eggenberger [3]
		Green [4]
		Menéndez [39]
		Nandhagopal [5]
		Rosini [33]
		Kumar [34]
		Sarwal [40]
		Shin [41]
		Uysal [42]
		Xie [43]
		Wondergem [44]
	Cavernoma (7%)	Maier [25]
		Perković [26]
	Intracranial capillary telangiectasia (3%)	Li [27]
	Hemorrhage (3%)	Xia [28]
Multiple sclerosis (17%)		Cárdenas-Rodríguez [2] Guler [6]
		Mortzos [29]
		Skaat [45]
		Wanono [46]
Tuberculosis (7%)		Shao [7]
		van Toorn [30]
Metastases (3%)		Ortiz-Pérez [8]

## 4. Conclusions

The eight-and-a-half syndrome is a rare neuro-ophthalmological condition due to lesions in the caudal pons, associated with a broad set of causes. Hence, it is essential to recognize the presence of symptoms and signs suggesting brainstem involvement, to address the most accurate diagnostic workup and appropriate therapy as soon as possible. Our review identified cerebrovascular disease as the most common cause of the eight-and-a-half syndrome, followed by demyelinating disorders. Besides, more peculiar etiologies such as atypical tuberculoma localizations and brain metastases (often rising from lung carcinoma) must be also taken into account. After describing the anatomy of the pathways controlling oculomotion, we have paid particular attention to the differential diagnoses from which the eight-and-a-half-syndrome must be differentiated. The prompt recognition of these disorders is critical to start a potential treatment as soon as possible, to avoid or limit sequelae. Indeed, due to the anatomical complexity of this region, time is critical to limit functional damage or life-threatening scenarios.

## Figures and Tables

**Figure 1 brainsci-12-00451-f001:**
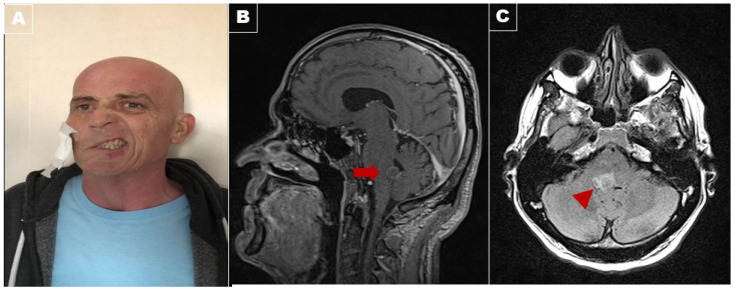
Panel (**A**): The patient showed severe right facial palsy with a peripheral pattern with Bell’s sign, rightward conjugate gaze palsy, and right internuclear ophthalmoplegia. The leftward gaze evoked left-beating nystagmus. Postcontrast T1-weighted Magnetic resonance imaging in the sagittal view (Panel (**B**)) demonstrated a ring enhancement lesion (arrow) on fluid-attenuated inversion recovery (FLAIR) in the axial view (arrowhead; Panel (**C**)) in the dorsal tegmentum of the pons, suggestive of metastasis.

## Data Availability

Data are available upon request.

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
