# Peer review of "A Rare Neuro-Ophthalmological Condition in a Patient with Lung Adenocarcinoma: The Eight-and-a-Half Syndrome, Case Report and Review of the Literature"

_brainsci, 2022, doi:10.3390/brainsci12040451_

Round 1

Reviewer 1 Report

In this interesting report, the authors present a rare case of eight-and-a-half syndrome due to a space-occupying lesion in the dorsal pons suspected to be a brain metastasis of an adenocarcinoma. Several major concerns should be addressed before this manuscript may be reconsidered:

Introduction: There are currently no references in the introduction section. The scientific soundness of the introduction section will certainly benefit from citing the appropriate literature.

Case report: Clinical information on follow-up is missing.

Discussion: The eight-and-a-half syndrome has been described first in 1999 by Eggenberger et al.. It therefore appears appropriate to refer to Eggenberger et al. (and not only to Bocos-Portillo et al.) at the first mentioning of the syndrome in the text. Reference #42, which is the only publication to date reporting on a metastatic lesion underlying the eight-and-a-half syndrome, has not been authored by Marimon-Suñol et al., but by Ortiz-Pérez et al.. This has to be corrected in table 1, and in the list of references.

Figure 2: The legend refers to a round shadow, while there is no round shadow in the figure.

Reviewer 2 Report

Introduction

1) In introduction authors have not provided sufficient information about Eight and a half syndrome 

Discussion and conclusion

Both are too long, there are some unnecessary information can be safely removed, Furthermore some main points such as association between adenocarcinoma, metastases and eight and a half syndrome or other brain stem disorder has not been well discussed. Please check references  again that are missed in sentences. 

Table

In the table for a good style, please summarize study for instance "Sampath Kumar N. S. et al., 2014 [25]" to Kumar [25]

In the heading of the table, you can change "case" to "Reference or study"

Image

For a good privacy, I think you can cover the eyes of patient (even though you "Informed Consent Statement: Written informed consent has been obtained from the patient to publish this paper"

Do you have content to publish his full image too please let us know? 

Round 2

Reviewer 1 Report

The reviewer's concerns have been addressed adequately.

Author Response

Thanks again for your time and valuable help